# *Borrelia burgdorferi* 0755, a Novel Cytotoxin with Unknown Function in Lyme Disease

**DOI:** 10.3390/toxins16060233

**Published:** 2024-05-21

**Authors:** Sam T. Donta

**Affiliations:** 1Department of Medicine, Division of Molecular Medicine and Division of Infectious Disease, Boston University Medical Center, Boston, MA 02118, USA; sdonta@comcast.net; Tel.: +1-508-548-5300; Fax: +1-508-540-0133; 2Department of Medicine, Falmouth Hospital, 100 Ter Heun Drive, Falmouth, MA 02540, USA

**Keywords:** Lyme disease, *Borrelia burgdorferi*, neurotoxin, Z ribonuclease

## Abstract

The pathophysiology of Lyme disease, especially in its persistent form, remains to be determined. As many of the neurologic symptoms are similar to those seen in other toxin-associated disorders, a hypothesis was generated that *B. burgdorferi*, the causative agent of Lyme disease, may produce a neurotoxin to account for some of the symptoms. Using primers against known conserved bacterial toxin groups, and PCR technology, a candidate neurotoxin was discovered. The purified protein was temporarily named BbTox, and was subsequently found to be identical to BB0755, a protein deduced from the genome sequence of *B. burgdorferi* that has been annotated as a Z ribonuclease. BbTox has cytotoxic activity against cells of neural origin in tissue culture. Its toxic activity appears to be directed against cytoskeletal elements, similar to that seen with toxins of *Clostridioides difficile* and *Clostridioides botulinum*, but differing from that of cholera and *E. coli* toxins, and other toxins. It remains to be determined whether BbTox has direct cytotoxic effects on neural or glial cells in vivo, or its activity is primarily that of a ribonuclease analogous to other bacterial ribonucleases that are involved in antibiotic tolerance remains to be determined.

## 1. Introduction

Lyme disease is caused by the spirochete *Borrelia burgdorferi*, transmitted by the bite of an infected *Ixodes* sp. tick [1]. Many of the symptoms of Lyme disease, especially in its persistent form, relate to the nervous system, including pain, paresthesias, facial nerve palsy, tremors, cognitive dysfunction, emotional lability, and fatigue [2]. The mechanisms responsible for persistent symptoms remain to be delineated. Some of the symptoms are similar to those seen in zoster (i.e., pain), and in tetanus (i.e., hypersensitivity to sensory stimuli, increased beta-adrenergic sensitivity) [3].

Studies in monkeys and dogs affirm that the major localization of the borrelia organisms is in the nervous system, both centrally and peripherally [4,5,6]. Using PCR, Pachner et al. [5] were able to detect the presence of *B. burgdorferi* DNA in the brains of chronically infected monkeys. In monkeys infected with *B. burgdorferi* for 46 months, Roberts et al. [4] demonstrated lesions in the peripheral nervous system characterized by both a loss of axons and a concurrent regeneration of the nerves. There was no indication of demyelination in those lesions. The lack of demyelination and the loss of axons implicate a possible toxin in the pathogenesis of the disease. The mouse model of Lyme disease reflects the immune response seen in some patients, but does not manifest the many other varied symptoms of the disease [7]. It has not yet been demonstrated whether the organisms exist in glial, neural, or endothelial cells, extracellularly or intracellularly. In vitro, borrelia organisms can invade and persist in neural cells [8], fibroblasts [9], and endothelial cells [10] for short periods of time, but the relevance of these observations to the in vivo state has yet to be established.

Bacterial toxins have been associated with alterations in the normal physiology of several organ systems. Diphtheria toxin ADP-ribosylates elongation factor 2, inhibiting protein synthesis [11]. Cholera and the related *E. coli* enterotoxins act as ADP-ribosyltransferases, targeting the Gs protein of the adenylate cyclase system, to increase cAMP [12]. These toxins bind to GM1 gangliosides, which are also abundant as sialoglycolipids in neuronal membranes [13]. Tetanus and botulinum neurotoxins bind to other membrane glycolipids in the central nervous system [14], with subsequent neurotoxic effects on calcium-dependent neurotransmitter release. Tetanus toxin acts as a zinc binding metalloendoprotease to cleave synaptobrevin 2, an integral protein in the fusion of synaptic vesicles with the plasma membrane of nerve cell synapses [15]. It therefore seemed reasonable to hypothesize that some of the alterations seen in neuroborreliosis are due to toxin-related events.

Some bacterial toxins target the cell’s cytoskeletal components, disrupting the normal functions of the cell. Botulinum C2 toxin modifies monomeric G-actin via ADP-ribosylation [16]. The ribosylation prevents the polymerization of actin by acting as a capping protein, leading to the depolymerization of actin filaments. In contrast, botulinum C3 exoenzyme ADP-ribosylates GTPase Rho, inactivating it, and causing a disruption of the actin cytoskeleton [17]. *C. difficile* toxin B uses an alternate mechanism to disrupt the cytoskeleton [18]; it glycosylates the protein RhoA, inactivating it and disrupting the actin filaments. The actions of these toxins on the cytoskeleton can be observed by their abilities ability to cause a rounding of cells in tissue culture. As the same effect was observed in tissue-cultured cells treated with BbTox, the subject of this report, it seems reasonable to hypothesize that some of the alterations seen in neuroborreliosis are due to toxin-related events.

## 2. Results

Initial experiments examined whether *B. burgdorferi* secretes a toxin when grown in culture. Media and bacterial lysates from 6- and 10-day growths, representing log and stationary phases, respectively, were used to assess toxin activity. Two assays were used to measure ADP-ribosyltransferase activity:, one detecting the ribosylation of G protein, a standard assay for cholera toxin activity [19] and the second detecting the ribosylation of elongation factor 2 (EF2), an assay of diphtheria toxin activity [20]. Activity was detected in both assays, albeit at low levels.

Having detected ribosyltransferase activity, the assumption was made that a borrelial toxin would contain some conserved sequences with known ADP-ribosylating toxins. PCR and degenerate primers were then designed to target regions of highly conserved amino acid sequences of various ADP-ribosylating toxins [21]. The amino acid sequences chosen are shown in Figure 1. The toxins were divided into two groups (1-cholera, *E. coli*, pertussis, and 2-diphtheria, pseudomonas) to prevent the primers from becoming too degenerate in nucleotide sequence. Primers for each group of toxins were designed to match highly conserved amino acid sequences that form the active site in the catalytic domain.

To search for toxins of *B. burgdorferi*, genomic DNA from strain 2591 was extracted for use as a template for PCR. No PCR products were detected using primers for the cholera/*E. coli*/pertussis toxin group. An amplification product of 600 bp was detected; however, this was using primers for the diphtheria/pseudomonas toxin group (Figure 2).

This product was cloned, and the DNA sequences were compared to the published *Borrelia burgdorferi* genome database [22]. The DNA of the cloned product was an exact match of the DNA sequence of a hypothetical *B burgdorferi* protein (BB0755) of then unknown function (Figure 3). Of note, this borrelial protein BB0755 is also conserved with a protein of unknown function in the spirochete *Treponema pallidum*.

Having the benefit of the genome database, the full-length coding region for BB0755 was amplified, using synthesized primers to the 5′ and 3′ ends of the coding region. BamH1 and HindIII sites in the primers were used to insert the amplified DNA into the linker site of a pET 30a expression vector (Novagen). The putative toxin was then expressed and purified, and named BbTox. Each stage of the induction and purification was examined by Western blot. Following purification, a protein of the expected size of 37 kD was obtained (Figure 4).

When the purified BbTox protein was tested in ADP-ribosylation assays, no activity was detected in the EF-2 assay indicating that BbTox was not responsible for the low levels of activity seen using bacterial lysates and culture media. This was not a surprising finding, as the primer sequence in BbTox is conserved at the DNA level with diphtheria toxin, but, interestingly, it is one nucleic acid out of frame with the amino acid sequence. In contrast, ADP-ribosyltransferase activity was detected in the agmatine ribosylation assay that measures cholera toxin-like activity, albeit an activity approximately one-fifth of that of the cholera toxin (Figure 5). This assay does not distinguish whether BbTox is intrinsically less active than cholera toxin or whether it has an endogenous target differing from that of cholera toxin.

Purified BbTox was then examined for possible activity in two tissue -culture model systems. Using Y1 mouse adrenal cells that detect cholera activity [12], BbTox caused a morphologic change (i.e., rounding) in the cells similar to that seen with cholera toxin and *C. difficile* toxin (Figure 6A).

The toxin’s activity was not manifested until 24 h of incubation, a time course much slower than that for cholera toxin. Treatment of C6 rat glial cells with 200 ng/mL of BbTox resulted in rounding of these cells within 10 to 18 h (Figure 6B).

As C6 cells are not responsive to cholera toxin, but do respond to *C. difficile* toxin with a similar morphologic response, BbTox would be acting by a mechanism differing from cholera toxin, more like a *C. difficile* toxin acting on the cytoskeleton. Antisera against cholera toxin and *C. difficile* toxin were without effect on BbTox, distinguishing BbTox from either of these other two toxins. Bacterial lysates and culture supernatants were without effect on either tissue -culture system. As any potential contaminants such as endotoxin are without effects on these tissue -culture cells, the cytotoxic activity noted with BbTox is unique.

C6 glial cells responded to BbTox in a dose- and time-dependent manner (Table 1). Cells treated with 100 ng or 200 ng of BbTox were 50% rounded by 8 h, and were 100% rounded by 36 h post-exposure to the toxin. Cells treated with amounts of toxin as low as 20 ng of BbTox were also effective, but the effects were not manifest until 5 days had elapsed. Interestingly, 2 ng/mL of BbTox was without effect. Similar dose responses were seen with Y1 adrenal cells.

Brefeldin A, an inhibitor of Golgi formation, has been of value in distinguishing toxins according to their intracellular trafficking patterns [23]. When C6 or Y1 cells were pretreated with Brefeldin A, the effects of BbTox were accelerated (Table 2), effects opposite that seen with the cholera/heat-labile (LT) *E. coli* enterotoxin group and Shiga toxins; in contrast, diphtheria toxin and the *C. difficile* toxins are unaffected by Brefeldin A [23]. These results further support for the hypothesis that BbTox is a novel toxin.

## 3. Discussion

Lyme disease is an infection that involves multiple tissues, including the peripheral and central nervous systems. The mechanisms responsible for the various clinical symptoms have not been delineated, with speculation and hypotheses involving persisting infection, lipoprotein-associated inflammation, and auto-immunity. As the manifestations of an increasing number of infectious diseases are being found to be associated with toxins, and as a number of the neurological symptoms of Lyme disease seem compatible with toxin-induced effects, it was speculated that the causative agent, *Borrelia burgdorferi*, might produce a neurotoxin to account for those effects. Reasoning that any such toxin might resemble already known toxins, degenerate primers were created to the active sites of several different groups of toxins. One of the amplification products of the diphtheria-pseudomonas toxin primer group was identical in sequence to a gene coding for a protein of *Borrelia burgdorferi* of then unknown function. This finding of having identities to the diphtheria-pseudomonas toxin primers, but being one nucleic acid out of frame, further supports the hypothesis that BbTox is a novel cytotoxin. This 37 kd protein was isolated as a recombinant protein and was cytotoxic to Y1 adrenal cells and C6 glial cells, both cells of neuronal origin, in tissue culture. The toxin, named BbTox, appears to be novel in that it displays effects similar to several different toxins, but is different from each of them. BbTox has a cholera-like morphologic effect on Y1 adrenal cells, and is capable of ADP-ribosylation of agmantine, a synthetic target of cholera toxin, although its ribosylating activity is much less than that of cholera. Also, in contrast to the cholera/heat-labile *E. coli* enterotoxin group of toxins, BbTox causes morphologic effects on cholera-resistant C6 rat glial cells, and its effects are not neutralized by antibodies to cholera toxin. The morphologic effects of BbTox resemble those of the *C. difficile* toxins, appearing to affect the cytoskeleton, with minimal effects on cell viability. In contrast to the *C. difficile* toxins, though, the effects of BbTox are not neutralized by antibodies to the *C. difficile* toxins. BbTox is also affected by Brefeldin A, an inhibitor of Golgi formation that does not affect either *C. difficile* toxins or diphtheria toxins. In contrast to other toxins that are affected by Brefeldin A, ii.e., pseudomonas exotoxin A, the cholera/*E. coli* enterotoxins, and Shiga toxin, the effects of BbTox are accelerated, and not inhibited. 

Based on observations thus far, BbTox appears to be a novel toxin whose origins and role in Lyme disease remain to be determined. It seems reasonable then, using the examples of tetanus and botulinum neurotoxins whose effects can last for weeks, to hypothesize that BbTox impairs one of the normal neurophysiologic processes. Whether the toxin is secreted as an exotoxin or whether the organism and the toxin occupy an intracellular reservoir are questions that need to be answered before there is a better understanding of the potential role of BbTox in Lyme disease. Peters and Benach [24] demonstrated that *B. burgdorferi* was toxic to cultures of differentiated PC12 rat neural cells. The effects of BbTox and the borrelia themselves in culture systems strengthen the hypothesis of toxin-induced symptoms of the disease. The *B. burgdorferi* spirochete can only rarely be found following the initial infection and dissemination, suggesting an intracellular reservoir for persistent infection and/or low numbers of bacterial infections. Studies in the Macaque model affirm that the major localization of the borrelia organisms is in the nervous system, both centrally and peripherally [5]. The exact location of the organisms has not yet been defined, whether extracellularly or intracellularly, in glial, neural, or endothelial cells. In vitro, borrelia organisms persist in endothelial cells [8] and fibroblasts [9] for short periods of time, but the relevance of these observations to the in vivo situation is unclear.

The entire genome of *Borrelia burgdorferi* has been published by The Institute for Genomic Research (TIGR) [22]. The sequencing of the genome identified a possible function for 50% of the genes. No new virulence factors were identified, and the only toxins identified were a family of hemolysins. What is currently known of *Borrelia burgdorferi* cannot explain the mechanism of action of the bacteria or the symptoms associated with Lyme disease. The symptoms of Lyme disease resemble many of the symptoms caused by other toxins. This suggests that *Borrelia burgdorferi* might produce a toxin that could account for some or all of the symptoms of Lyme disease, in particular neuroborreliosis. Further study of BbTox may help us to better understand the neuropathology of the disease, and possibly find new applications for the diagnosis and treatment of Lyme disease.

Since our initial identification of BbTox as a putative neurotoxin, BB0755 has been annotated as a Z ribonuclease [25]. Z ribonucleases are primarily found in *Trepomena* sp. and *Borrelia* sp., distinguishing them from other microbial ribonucleases. As BbTox is a Z ribonuclease, it remains to be determined whether it is an endoribonuclease or exoribonuclease. M ribonucleases have been identified as having internal toxin–-antitoxin systems involved in antibiotic tolerance and the persistence of pathogens [26,27,28]; thus, it remains to be determined if any other Z or other ribonucleases can have a similar function. Also to be determined is whether any other ribonucleases have the same effects as BbTox on tissue-cultured cells. Whether BbTox acts in vivo as a neurotoxin or its primary role is in creating an antibiotic-tolerant state that promotes the persistence of the bacteria are important questions for further study.

The justification for naming BbTox (BB0755) a toxin is based on its unique activity in specific tissue-culture cell systems, where morphologic changes similar to those observed with BbTox appear to be specific to toxins thus far identified to produce such effects, and not attributable to non-specific effects induced by any other known substances. If the tissue-culture effects observed in these studies can be replicated by other non-spirochetal ribonucleases that are part of toxin—antitoxin systems involved in the persistence of other antibiotic-tolerance microbes, then those findings could have a considerable impact on the management of patients with persistent Lyme disease and other antibiotic-tolerant infections.

## 4. Materials and Methods

### 4.1. Bacterial Strains and Plasmids

Strain 2591 of *B. burgdorferi*, the same strain used by Padula et al. [29], was grown in BSK H medium with 6% rabbit serum (Sigma, Kawasaki, Japan) at 34 °C under microaerobic conditions. Borrelia DNA was collected from bacteria, resuspended in TE buffer, and boiled for 10 min. PCR products were cloned into a pET 30 (Novagen, Metro Manila, Philippines) vector, and the sequences were confirmed. Plasmids were transformed and propagated in *E. coli* strain NovaBlue. The host strain used for protein expression was *E. coli* BL21(DE3).

The pET30 expression plasmid contains a T7lac promoter, the kanamycin resistance gene, and two N-terminally fused tags. The first is the His-Tag sequence, encoding six histidine residues, enabling the purification of target proteins by metal chelation chromatography. The second encodes the 15 amino acid S-Tag peptide for the detection of target proteins by Western blot analysis. 

### 4.2. Expression and Purification of Toxins

Target proteins carrying the N-terminal histidine tag and S-protein tag were expressed and purified as described by the manufacturer (Novagen). In brief, gene expression was driven by T7 polymerase to induce the expression of target proteins. Transformed *E. coli* BL21(DE3) were grown in LB plus kanamycin at 37 °C until A600 = 1.0, when IPTG (1 mM) was added. Consecutive 1 h time point samples were taken to check for protein induction with SDS-PAGE. The bacteria were then harvested by centrifugation and frozen overnight at −20 °C. To isolate the toxins as insoluble inclusion bodies, the bacteria were repeatedly sonicated and washed in His-Bind Binding buffer (5 mM imidazole, 0.5 M NaCl, 20 mM Tris-HCl, pH 7.9). To prevent protein degradation, 1/200 volume of 0.2 M PMSF was added to the buffers. The proteins were then solubilized in 6 M guanidine.

Using the His-Bind resin and buffer kit, the expressed toxins were purified under denaturing conditions by metal chelation chromatography. The resin was charged with 50 mM NiSO_4_. In the presence of a binding buffer plus 6 M guanidine, the N-terminal histidine residues bind to the nickel cations in the resin. All contaminating proteins were washed off the column. The toxins were then recovered in Elute buffer (0.5 M imidazole, 0.5 M NaCl, 20 mM Tris-Hcl, pH 7.9, 6 M guanidine). The collected purified proteins were refolded by dialysis in a Spectra/Por membrane, 12–14,000 MW cut off against a rRefolding buffer (50 mM Tris pH 8.0, 50 mM NaCl) with decreasing amounts of guanidine-HCl with each successive change of buffer. The refolded purified toxins were stored at 4 °C.

### 4.3. SDS-PAGE and Immunoblotting

The mini-Protean II gel apparatus (Bio-Rad, Hercules, CA, USA) and 10% polyacrylamide gels were used to analyze the proteins. Proteins were stained with Coomassie Brilliant Blue. Western blotting was performed using the Semiphore transfer unit (Hoefer Pharmacia, Holliston, MA, USA), Immobilon-P membrane (Sigma), and the S-Tag Western Blot kit. The S-protein tag fused to the recombinant toxins has a high-affinity interaction with ribonuclease S-protein. Blots were transferred, blocked with 1% gelatin, incubated with S-protein alkaline phosphatase conjugate, and then detected colorimetrically with the addition of alkaline phosphatase buffer plus nitro blue tetrazolium and 5-bromo-4-chloro-3-indolyl phosphate.

### 4.4. Ribosylation Assay

The agmatine ribosylation assay was used to measure cholera toxin ribosyltransferase activity [19]. Samples collected from the bacteria were denatured in a reaction mix of 50 mM potassium phosphate, pH 7.5, and 20 mM DTT for 20 min at 30 °C. To this mixture were added ovalbumin (0.3 mg/mL), GTP (1 mM), agmatine (20 mM), and 14-C-NAD (5 nM). The reaction was incubated for 60 min at 30 °C and immediately passed over equilibrated 2 mL bed volume AG1-X2 columns (BioRad)and eluted with 20 mL in 2 mL fractions. Collected fractions were assayed for the cpm of radioactivity in a beta-scintillation counter (Beckman, Brea, CA, USA). 

### 4.5. Tissue Culture

All cell lines were maintained as monolayer cultures at 37 °C in a humidified atmosphere of 5% CO_2_. Y1 (mouse adrenal tumor) and C6 (rat glial) tissue-culture cells were propagated and maintained in Ham’s F10 plus 15% horse serum and 2.5% fetal bovine serum. All cell media contained penicillin (100 units/mL), streptomycin (100 µg/mL), and glutamine (1 mM).

For cellular assays, 5 × 10^4^ cells/mL were seeded in 12-well plates. The medium was changed 18 h prior to the addition of the toxin. For morphologic studies, the cells were observed at a series of time points following the addition of the toxin. For cyclic AMP studies, media removed from the wells at these time points were evaluated for increased cAMP levels using the cAMP [125-I] RIA kit (NEN Life Science, Boston, MA, USA).

## Figures and Tables

**Figure 1 toxins-16-00233-f001:**
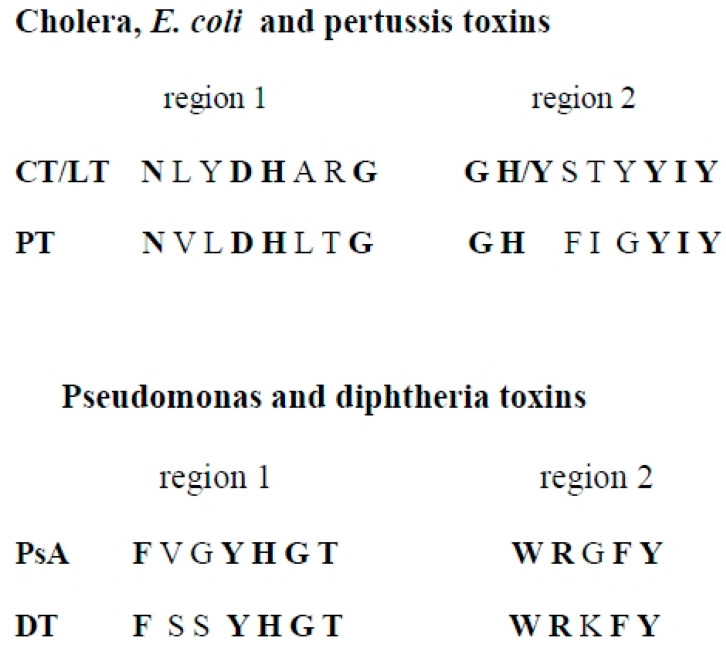
Amino acid sequences of the primers used in *Borrelia burgdorferi* PCR. Cholera, *E*. *coli*, pertussis, pseudomonas, and diphtheria primer sequences were derived from the conserved amino acid sequences that form the active site of the ADP ribosylating toxins.

**Figure 2 toxins-16-00233-f002:**
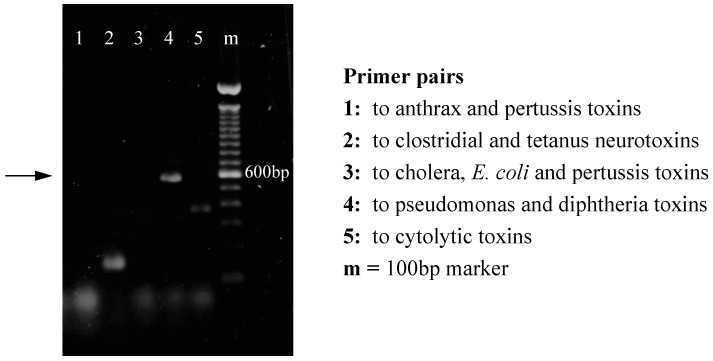
PCR detection of putative *B. burgdorferi* toxins using degenerate primers derived from known toxin groups. DNA extracted from *B. burgdorferi* was amplified by PCR using each pair of primers indicated. PCR products were visualized by electrophoresis on a 1% agarose gel.

**Figure 3 toxins-16-00233-f003:**
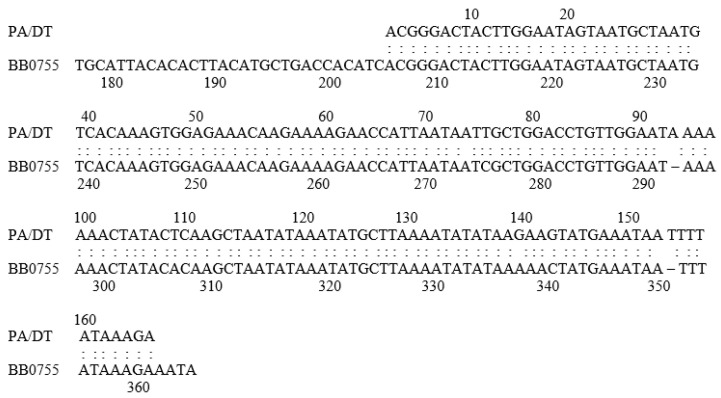
Comparison of sequences of amplification product and section of BB0755.

**Figure 4 toxins-16-00233-f004:**
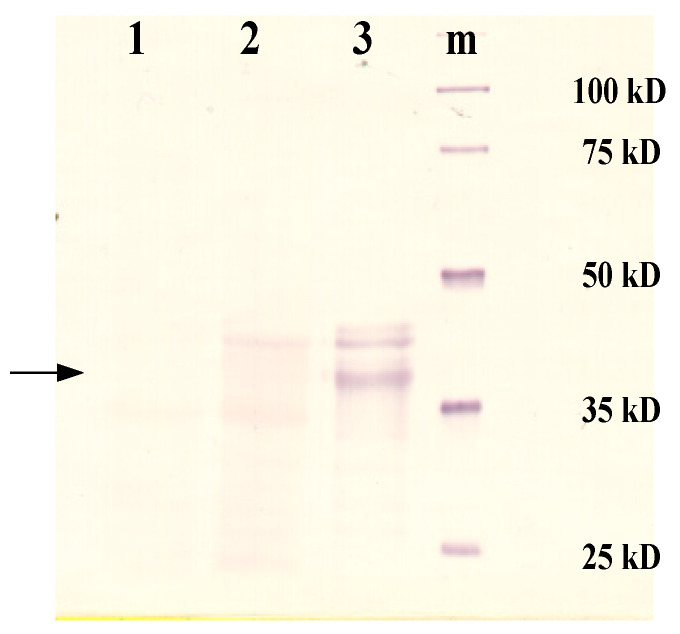
Western blot analysis of the expression and purification of Bb tox. Lane 1: pre IPTG induction of pET30/BbTox. Lane 2: post IPTG induction of pET30/BbTox. Lane 3: purified BbTox. Lane m: standard markers. Expected molecular weight of 37 kD is indicated by the arrow.

**Figure 5 toxins-16-00233-f005:**
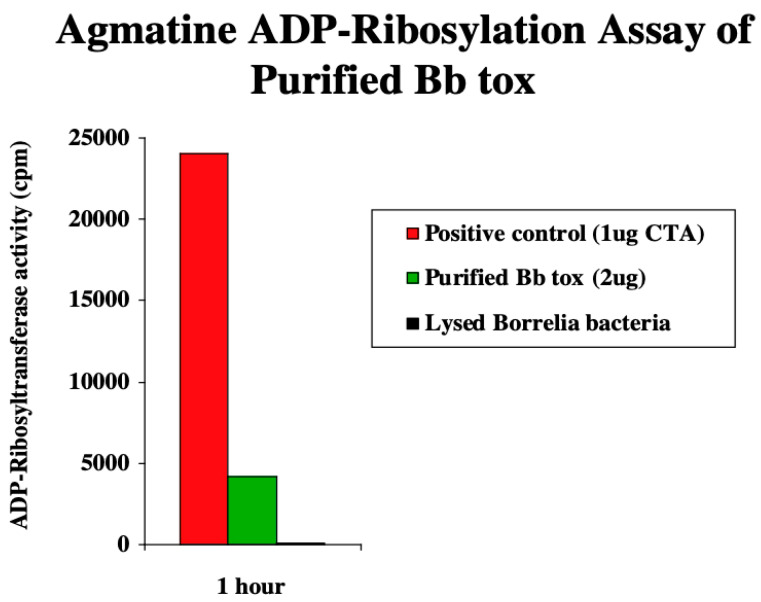
ADP-ribosyltransferase activity of BbTox. The activity of purified BbTox was compared to a sample of lysed *B. burgdorferi* (not significantly different from background) and to cholera toxin. The enzymatic activity is measured in cpm and corrected for background.

**Figure 6 toxins-16-00233-f006:**
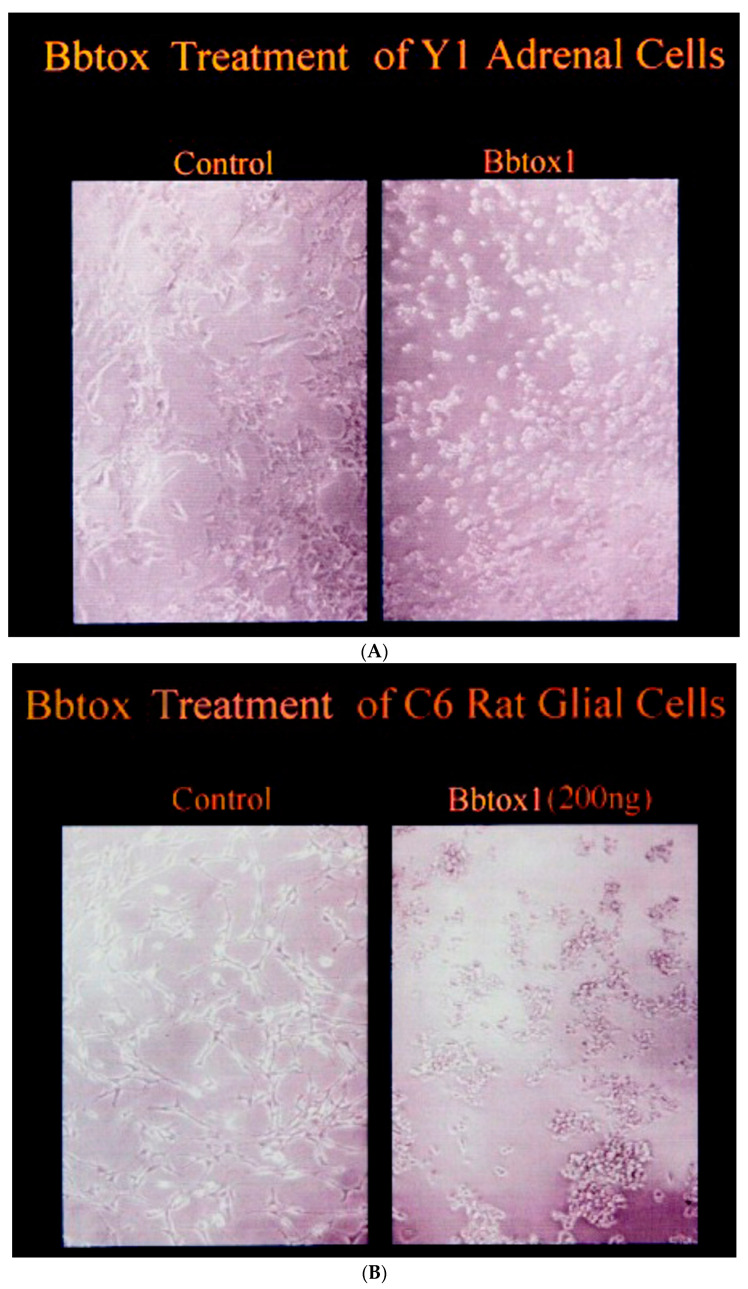
(**A**) Treatment of Y1 adrenal mouse cells with BbTox. (**B**) Treatment of C6 rat glial cells with BbTox.

**Table 1 toxins-16-00233-t001:** Morphological effects of BbTox in C6 glial cells. C6 rat glial cells were plated at 1 × 10^5^ per well 24 h prior to treatment with BbTox. Control wells were treated with Tris buffer. Morphological effects were measured as percentage of rounded cells per total cell number at 8 h, 36 h, and 5 days post-treatment.

	% Rounding of C6 Glial Cells		
Treatment	8 h	36 h	5 Days
Bbtox 200 ng	58	100	100
Bbtox 100 ng	51	100	100
Bbtox 20 ng	32	25	57
Control buffer	27	10	22

**Table 2 toxins-16-00233-t002:** Effect of Brefeldin A on the activity of BbTox. Y1 cells were pretreated for 1 h with Brefeldin A (BFA) 0.5 mcg/mL or with control buffer, then the effects of BbTox on rounding of Y1 cells noted after 24 and 48 h.

	% Rounding of Y1 Adrenal Cells			
Treatment	24 h		48 h	
	−BFA	+BFA	−BFA	+BFA
BbTox 200 ng	50	100	100	100
BbTox 20 ng	10	30	75	65
Control buffer	10	10	10	10

## Data Availability

The raw data supporting the conclusions of this article will be made available by the authors on request.

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
