# Peer review of "Borrelia burgdorferi 0755, a Novel Cytotoxin with Unknown Function in Lyme Disease"

_toxins, 2024, doi:10.3390/toxins16060233_

Round 1

Reviewer 1 Report

Comments and Suggestions for Authors

General Comments

The manuscript entitled “Borrelia burgdorferi 0755, a Novel Cytotoxin with Unknown Function in Lyme Disease” describes the first reported toxin associated with Borrelia burgdorferi (Bb), the spirochetal agent of Lyme disease. Previous studies using older techniques have failed to identify a toxin in Bb (Strnad et al, Virulence. 2023;14:2265015). The authors have done a good job of identifying and characterizing the toxin, which they call BbTox, and the manuscript merits publication. Several questions need to be addressed.

Major Comments

1. Introduction, line 37: The authors state that “It has not yet been demonstrated whether the organisms (Bb) exist in glial, neural, or endothelial cells, extracellularly or intracellularly.” However there is evidence that Bb can invade neural cells (Livengood et al, Infect Immun. 2008;76:298-307; Williams et al, PLoS One. 2018;13:e0197413) as well as fibroblasts and endothelial cells (Wu et al, Infect Immun. 2011;79:1338-48). The presence of a Bb toxin could therefore have profound effects on these diverse tissues.

2. Methods, line 67: Strain 2591 of Bb was used for the study. The authors should give some background about this North American strain (eg Padula et al, Infect Immun. 1993;61:5097-105). Do all species of Bb have BB0755, and does sequence variation occur in the genome product (Schwartz et al, Curr Issues Mol Biol. 2021;42:409-454)?

3. Line 119: Y1 (mouse adrenal tumor) and C6 (rat glial) tissue culture cells were used to test the virulence of the putative toxin. Were any other cell lines examined? Although Bb can cause neurological problems, musculoskeletal and cardiac symptoms can also occur and a toxin might influence these cell types.

4. Discussion: The authors are appropriately cautious about the effects of the putative Bb toxin. In particular the paradoxical response to Brefeldin A requires further investigation. Does BbTox induce cytokine secretion by macrophages (Strle et al, J Infect Dis. 2009;200:1936-43)?

5. Line 155: “BB0755 is also conserved with a protein of unknown function in the spirochete Treponema pallidum.” Does the syphilis spirochete have a known toxin?

Minor Comments

1. Introduction, line 26: The formal names of Herpes zoster and Clostridioides tetani should be used.

2. Introduction, line 39: “endothelial cells (8), fibroblasts, (9) and endothelial cells (10)”. References 8 and 10 can be combined.

3. Introduction, line 60: glycosylates

4: Figure 4: Lanes should be marked on the Western blot. The arrow is missing.

5: Discussion, Line 244: “BbTox is also affected by Brefeldin A, an inhibitor of Golgi formation that does not affect either C. difficile toxins or diphtheria toxin.”

6: Discussion, line 249: “It seems reasonable then, using the examples of tetanus and botulinum neurotoxins whose effects can last for weeks, to hypothesize….”

7. Reference 27: toxin-antitoxin

Author Response

I thank the reviewer for the important suggestions and have incorporated them into the revised manuscript. Specifically:

Reference to the study by Livengood has been added to the text.

The strain used in our study was further noted to be used by Padula et al.

Other species of B.burgdorferi were not examined to determine if all contain BB0755, nor if sequence variation occurs.

No other cell lines were examined. No studies were done to determine if cytokine secretion is affected in macrophages.

It's not known if Treponema palliidum has a known toxin.

All the minor comments were addressed.

Reviewer 2 Report

Comments and Suggestions for Authors

Line 43: good to cite Livengood & Gilmore article which demonstrates in vitro entry of Bb in to glial cells. Livengood JA, Gilmore RD Jr. Invasion of human neuronal and glial cells by an infectious strain of Borrelia burgdorferi. Microbes Infect. 2006 Nov-Dec;8(14-15):2832-40. doi: 10.1016/j.micinf.2006.08.014. Epub 2006 Sep 22. Erratum in: Microbes Infect. 2015 Jun;17(6):e1. PMID: 17045505.

Line 83-84: should this read something like "...were designed to match highly conserved.."  or "...correspond to highly conserved..."?

Line 98-99: better English usage to arrange as "...was detected, however, using primers..." (e.g. avoid a 'split infinitive'?)

Line 150 & also Line  223: Is there any pertinence, or is it purely a coincidence that Gilmore reports a 37 kDa band as being highly specific for Bb on Lyme Western blot? If pertinent, perhaps discuss? Gilmore RD Jr, Murphree RL, James AM, Sullivan SA, Johnson BJ. The Borrelia burgdorferi 37-kilodalton immunoblot band (P37) used in serodiagnosis of early lyme disease is the flaA gene product. J Clin Microbiol. 1999 Mar;37(3):548-52. doi: 10.1128/JCM.37.3.548-552.1999. PMID: 9986810; PMCID: PMC84463.

Line 241: "...the effects of which can last...:" probably better than "..which effects..."???

Lines 251-254: again Livengood & Gilmore speak to Bb within glial cells in an in vitro model.

Very interesting work which ought to stimulate further investigation in vitro, in animal models and in the clinical setting.

Donta had publicly hypothesized the possible presence of a toxin in Bb, probably in the early 1990s although at the time likely lacking the technology to further assess the question.  Not sure if the cited references by Donta include that speculation or not.  If not, it might be worth mentioning that he was (the Reviewer believes) the first physician/scientist to proffer that hypothesis.

Nice work. Congratulations.

Line 241: 

Comments on the Quality of English Language

Fine.  Some minor stylistic suggestions made to the authors

Author Response

I thank the reviewer for the important suggestions and have incorporated them into the text, including the article by Livengood et al.

Round 2

Reviewer 1 Report

Comments and Suggestions for Authors

Good job responding to reviewer comments. Further evaluation of BbTox may lead to novel treatments for Lyme disease.